# Absolute Accelerometer-Based Intensity Prescription Compared to Physiological Variables in Pregnant and Nonpregnant Women

**DOI:** 10.3390/ijerph17165651

**Published:** 2020-08-05

**Authors:** Philipp Birnbaumer, Pavel Dietz, Estelle Dorothy Watson, Gudani Mukoma, Alexander Müller, Matteo Christian Sattler, Johannes Jaunig, Mireille Nicoline Maria van Poppel, Peter Hofmann

**Affiliations:** 1Institute of Human Movement Science, Sport and Health, University of Graz, 8010 Graz, Austria; philipp.birnbaumer@uni-graz.at (P.B.); alexander.mueller@uni-graz.at (A.M.); matteo.sattler@uni-graz.at (M.C.S.); jaunig.johannes@uni-graz.at (J.J.); mireille.van-poppel@uni-graz.at (M.N.M.v.P.); 2Institute of Occupational, Social and Environmental Medicine, University Medical Center of the University of Mainz, 55131 Mainz, Germany; pdietz@uni-mainz.de; 3School of Therapeutic Sciences, Faculty of Health Sciences, University of the Witwatersrand, Johannesburg 2000, South Africa; estelle.owen@googlemail.com (E.D.W.); mukomagudani@gmail.com (G.M.); 4South African MRC/Wits Developmental Pathways for Health Research Unit, Faculty of Health Sciences, University of the Witwatersrand, Johannesburg 2000, South Africa

**Keywords:** ventilatory threshold, walking test, ENMO, raw accelerations, accelerometer cut point

## Abstract

Estimation of the intensity of physical activity (PA) based on absolute accelerometer cut points (Cp) likely over- or underestimates intensity for a specific individual. The purpose of this study was to investigate the relationship between absolute moderate intensity Cp and the first ventilatory threshold (VT_1_). A group of 24 pregnant and 15 nonpregnant women who performed a submaximal incremental walking test with measures of ventilatory parameters and accelerations from three different accelerometers on the wrist (ActiGraph wGT3X-BT, GENEActiv, Axivity AX3) and one on the hip (Actigraph wGT3X-BT) were analyzed. Cp were determined corresponding to 3 metabolic equivalents of task (MET), using the conventional MET definition (Cp_3.5_) (3.5 mL/kg×min) and individual resting metabolic rate (Cp_ind_). The ventilatory equivalent (VE/VO_2_) was used to determine VT_1_. Accelerations at VT_1_ were significantly higher (*p* < 0.01) compared to Cp_3.5_ and Cp_ind_ in both groups. Cp_3.5_ and Cp_ind_ were significantly different in nonpregnant (*p* < 0.01) but not in pregnant women. Walking speed at VT_1_ (5.7 ± 0.5/6.2 ± 0.8 km/h) was significantly lower (*p* < 0.01) in pregnant compared to nonpregnant women and correspondent to 3.8 ± 0.7/4.9 ± 1.4 conventional METs. Intensity at absolute Cp was lower compared to the intensity at VT_1_ independent of the device or placement in pregnant and nonpregnant women. Therefore, we recommend individually tailored cut points such as the VT_1_ to better assess the effect of the intensity of PA.

## 1. Introduction

Accelerometers are often used as a measure of free-living physical activity (PA) to estimate frequency, duration, and intensity [1]. In order to estimate the intensity of PA, calibration of accelerometers based on physiological responses to various activities is required. Calibration studies use oxygen consumption as the criterion measure and specific statistical methods, such as regression models or machine learning-based modeling, to determine the corresponding accelerometer cut point at a certain intensity threshold [2,3]. Usually, these intensity thresholds are used to define activity classes based on the average energy expenditure, expressed in multiples of the conventional metabolic equivalent of task (MET) ratio. The 1 MET value is defined as the energy expended by a subject at rest, which equals an oxygen consumption of 3.5 mL/kg×min for a 70 kg person [4]. Common classifications of PA intensity classes are light (< 3 METs), moderate (3–5.9 METs), and vigorous (≥ 6 METs) [5,6,7]. However, using the conventional 1 MET to classify PA intensity classes already causes misclassifications, compared to approaches where the individual resting metabolic rate would have been used [8,9]. Moreover, it must be considered that accelerometer cut points, which are based on the same absolute approach, are independent of individual performance capacity which leads to different physiological and metabolic strain (internal load) at the same absolute intensity [10]. To counteract that, relative accelerometer cut points based on the individual maximum performance capacity can be used. Several studies showed that the estimation of the duration of moderate-to-vigorous physical activity (MVPA) was shorter when relative comparisons to absolute cut points were applied, independent of body mass index (BMI). Therefore, absolute cut points overestimate MVPA compared to relative cut points [2,11]. When applying relative cut points, activity counts at a fixed relative intensity (e.g., 60% heart rate reserve (HRR)) increase with increasing fitness level [12]. On the contrary, low-fit persons had significantly higher percentage of maximum oxygen uptake (%VO_2max_) at absolute cut points (e.g., 2020 cpm) compared to fit persons [11]. The use of absolute accelerometer cut points is therefore likely to over- or underestimate PA intensity and volume in a certain PA intensity class for a specific individual. Individualized activity measurements (i.e., relative accelerometer cut points), which take into account the performance capacity of each individual, will allow us to draw more valid conclusions about the internal loads due to different intensities of PA.

Relative accelerometer cut points are usually derived using fixed percentages of an individual’s maximum heart rate (HR_max_), maximum oxygen uptake (VO_2max_), or HRR [13]. Such an individualized approach improves validity, but will still lead to some inaccuracy in the prescription of exercise intensity, because exercising at a calculated and fixed relative intensity such as 85% HR_max_ causes different metabolic and cardiorespiratory responses across individuals [14,15]. To overcome these problems, exercise can be prescribed based on submaximal markers such as the first and second ventilatory or lactate thresholds (VT_1_ and VT_2_; LTP_1_ and LTP_2_) which rely on the detection of physiological thresholds dependent on exercise intensity [16,17]. Hence, less variability in interindividual metabolic responses is expected when being active at a certain intensity relative to these thresholds. Indeed, Moser et al. [18] recently showed consistent metabolic responses during continuous cycle ergometer exercise five percent above and below LTP_1_ and LTP_2_. Furthermore, lactate thresholds as well as their physiological equivalents VT_1_ and VT_2_ allow conclusions about the fitness level and adaptations to exercise, and are very sensitive in reflecting differences in endurance performance [16,17].

Two studies by Gil-Rey et al. [19,20] estimated the intensity of PA from individually tailored accelerometer cut points derived from lactate thresholds in postmenopausal women. Individually tailored cut points revealed similar time for MVPA in high- and low-fit groups. In contrast, MVPA was overestimated in low-fit and more strongly in the high-fit group when absolute accelerometer cut points at moderate intensity (3–5.9 METs) were applied. They showed that individually tailored rather than traditional absolute accelerometer cut points estimate an individual’s activity level (i.e., time spent in different intensities of PA) more accurately. Thus, using individually tailored cut points from physiological thresholds may provoke greater adaptions to exercise and reduced interindividual variability of metabolic responses, as well as less overestimation of PA intensity. To date, it is unknown how well absolute accelerometer cut points are related to a physiological threshold such as the VT_1_ in pregnant as well as in young nonpregnant women. We hypothesized that the two absolute accelerometer cut points, derived from the 3 MET moderate intensity definition using either the conventional 3.5 mL/kg×min or the individual resting metabolic rate, will be lower compared to the individual threshold derived from VT_1_.Therefore, the aim of this study was to compare accelerometer cut points from different devices and placements, derived from the 3-MET absolute moderate intensity definition with the first ventilatory threshold (VT_1_) determined in a short submaximal incremental walking test (IWT) in a sample of pregnant and younger nonpregnant women.

## 2. Materials and Methods

### 2.1. Subjects

In total, a group of 30 pregnant (second and third trimester) and 17 nonpregnant women were tested at the University of Graz, Austria and the University of the Witwatersrand, Johannesburg, South Africa (22 pregnant, 9 nonpregnant). Subjects in Graz were active pregnant women and a mix of highly active and well-trained nonpregnant women (mainly students). In South Africa, subjects were sedentary and low-active pregnant women and a mix of sedentary, low-active, and well-trained nonpregnant women. Before any study procedures were undertaken, the participants completed informed consent forms and were familiarized with the testing protocol. The study protocol was approved by the local ethics committees (SA clearance certificate no. M160532; GZ. 39/42/63 ex 2015/16).

### 2.2. Test Protocol

Participants performed one IWT while wearing a portable gas analyzer and four different accelerometers. Before the IWT, resting metabolic rate was measured during 15 min of supine resting. IWT was conducted on a 400 m outdoor running track and started at 3 km/h. Walking speed was paced by audio signals given in 10 m intervals and increased by 0.5 km/h every 50 m up to the maximum individual walking speed. Maximum walking speed was defined as the speed where participants were unable to walk the given pacer speed.

### 2.3. Measurements

Gas exchange data were continuously measured breath by breath by a portable gas analyzer in Graz (CORTEX METAMAX 3B, Cortex Biophysik GmbH, Germany) and Johannesburg (OXYCON Mobile, CareFusion GmbH, Hoechberg, Germany). Calibration of volume, O_2_, and CO_2_ gas sensors was performed prior to every test according to the manufacturer’s guidelines. For activity measurements, all participants were equipped with four accelerometers in total. Three different types of accelerometers were attached on the nondominant wrist, placed in a random order: ActiGraph wGT3X-BT (ActiGraph, Pensacola, FL), GENEActive (Activeinsights, Kimbolton, UK), Axivity AX3 (Axivity Ltd., Newcastle upon Tyne, UK). In addition, one accelerometer (ActiGraph wGT3X-BT) was placed on the left hip. Prior to the measurements, all accelerometers were initialized with a data sampling frequency of 100 Hz and a sampling range of ± 8 g.

### 2.4. Determination of Physiological Thresholds and Accelerometer Cut Points

Gas exchange data were exported into Microsoft Excel files (Microsoft Corporation, Redmond, WA, USA) in 15 s epochs using the manufacturer’s software. Oxygen uptake (VO_2_) was converted to MET_3.5_ by using the conventional conversion factor (1 MET = 3.5 mL/kg×min) and to MET_ind_ by using the individual resting metabolic equivalent (1 MET_ind_) defined as mean oxygen uptake from the last 10 min of the 15 min supine resting position. Based on the raw triaxial accelerations, the vector magnitude (expressed in milligravity (mg) units) was calculated using the Euclidian norm minus one (ENMO = √(a_x_^2^+a_y_^2^+a_z_^2^)-1g) [21]. Therefore, the raw files from all devices were imported into R statistical software V3.1.2 (R Foundation for Statistical Computing, Vienna, Austria) by which the metric ENMO was calculated in 15 s epoch using the package GGIR V1.2-0. Processed files were then exported into Microsoft Excel. To determine the absolute accelerometer cut points at moderate intensity (3 METs), we performed an individual linear regression analysis between the ENMO and the oxygen uptake during the IWT, based on MET_3.5_ (Cp_3.5_) and MET_ind_ (Cp_ind_). Individual physiological threshold was defined as the first ventilatory threshold (VT_1_), using the ventilatory equivalent (VE/VO_2_) to determine VT_1_ as the minimum of VE/VO_2_ without an increase in VE/VCO_2_. The evaluation was carried out by two independent examiners using computer-supported linear regression analysis to increase objectivity. In case of disagreement, the results were discussed with a further examiner.

### 2.5. Statistical Analysis

Data analysis was performed using GraphPad Prism 7 (GraphPad Software, San Diego, CA, USA). The Shapiro–Wilk test was used for confirmation of normality. For normally distributed data, independent t tests were used to assess differences between pregnant and nonpregnant women. If data were not normally distributed, Mann–Whitney U tests were applied. To determine the effect of different cut points (Cp_3.5_, Cp_ind_, and VT_1_) and devices/placement (GENEActiv, Axivity, ActiGraph wrist and hip) on acceleration (ENMO), we applied a two-way repeated measures ANOVA (two within subject factors) with post hoc Tukey multiple comparison test in pregnant and nonpregnant women. If sphericity was violated, the Geisser–Greenhouse correction was used. Spearman correlation coefficient was applied to evaluate the relationship between walking speed and accelerations at VT_1_. Data are presented as means ± standard deviation (M ± SD). Statistical significance was set at *p* < 0.05.

## 3. Results

We analyzed 39 data sets of healthy pregnant (*n* = 24; 27.7 ± 4.6 yrs) and nonpregnant (*n* = 15; 24.3 ± 2.2 yrs) women. In total, eight tests were excluded from the analysis because of incomplete data sets (six in the pregnant and two in the nonpregnant group). The mean age, weight, and BMI were significantly higher in pregnant women (gestational age: 26 ± 7 weeks), but maximum walking speed (v_max_), VO_2max_, speed at VT_1_ (v_VT1_), and MET_ind_ were significantly lower compared to nonpregnant women. Absolute oxygen uptake at VT_1_ (VO_2VT1_) was not different (*p* = 0.07) between pregnant (0.9 ± 0.2 L/min) and nonpregnant (1.0 ± 0.2 L/min) women, but calculated conventional MET values at VT_1_ were significantly lower in pregnant compared to nonpregnant women (3.8 ± 0.7 vs. 4.9 ± 1.4 METs, *p* < 0.01). Determined 1 MET_ind_ was significantly higher compared to the conventional 1 MET value (3.5 mL/kg×min) in pregnant and nonpregnant women (Table 1). Bland–Altman interobserver comparison of VT_1_ determination (VO_2VT1_ bias: 0.01 ± 0.08 L/min) revealed a high level of agreement for the analysis.

In general, accelerometer cut points (ENMO) showed higher values and higher interindividual variability in pregnant compared to nonpregnant women (e.g., SD = 136 vs. SD = 50 for Cp3.5). In both groups, interindividual variability was less for the hip worn device (Table 2). Two-way repeated measures ANOVA in pregnant woman showed a significant effect for the determination of cut points (F (1.76, 141.30) = 51.27, *p* < 0.01) and a significant effect for devices/placement (F (3, 80) = 3.16, *p* < 0.05) and no significant effect of interaction (F (6, 160) = 0.35, *p* = 0.91). Post hoc multiple comparison revealed no difference between Cp_3.5_ and Cp_ind_, but ENMO was significantly lower for Cp_3.5_ and Cp_ind_ compared to the ENMO at VT_1_ in pregnant women. Comparison of devices in pregnant women showed significantly higher ENMOs for wrist-worn GENEActiv compared to hip-worn ActiGraph (*p* < 0.05) and for wrist-worn ActiGraph compared to hip-worn ActiGraph and wrist-worn Axivity (*p* < 0.05). In nonpregnant women, there was a significant effect for the determination of cut points (F (1.28, 69.36) = 56.19, *p* < 0.01) and no significant effect for devices/placement and interaction (F (3, 54) = 0.42, *p* = 0.74; F (6, 108) = 0.38, *p* = 0.88). ENMO was significantly different for Cp_3.5_ and Cp_ind_ compared to the ENMO at VT_1_ and between Cp_3.5_ and Cp_ind_ (*p* < 0.01).

Figure 1 shows the oxygen uptake expressed in MET_3.5_ and MET_ind_ as well as the ENMO of all devices at comparable walking speeds of the IWT for pregnant and nonpregnant women. Walking speed and accelerations at VT_1_ were not significantly correlated in pregnant (r_preg_) but significantly correlated in nonpregnant (r_non_) women for GENEActiv (r_preg_ = 0.13/r_non_ = 0.57) and Axivity (r_preg_ = 0.30/r_non_ = 0.69). Both groups showed no significant correlation between v_VT1_ and ENMO at VT_1_ for wrist-worn ActiGraph (r_preg_ = 0.10/r_non_ = 0.41), but for hip-worn ActiGraph, values were significantly correlated (r_preg_ = 0.62/r_non_ = 0.69). Comparing ENMOs within all wrist-worn devices, GENEActiv and Axivity were similar in their measurements while wrist-worn ActiGraph showed higher mean accelerations with increasing speed. This difference was stronger in pregnant compared to nonpregnant women and at higher speeds above VT_1_. Walking speed at VT_1_ corresponded to 3.8 ± 0.7 and 4.9 ± 1.4 conventional METs in pregnant and nonpregnant women, respectively. Values ranged between 2.48 and 7.73 conventional METs and were lower at VT_1_ compared to the 3-MET absolute moderate intensity definition in three cases in pregnant and in one case in nonpregnant women.

## 4. Discussion

Accelerometer cut points at absolute moderate intensity definition (3 METs) were significantly lower compared to the intensity at VT_1_ in a short maximal incremental walking test. The underestimation of intensity compared to VT_1_ was independent of the accelerometer device or placement and the applied 1 MET value in pregnant and nonpregnant women. Walking speed at VT_1_ was 5.7 ± 0.5 and 6.2 ± 0.8 km/h, which corresponded to an oxygen uptake of 3.8 ± 0.7 and 4.9 ± 1.4 conventional METs in pregnant and nonpregnant women, respectively. Whether during pregnancy or not, a certain duration of moderate-intensity physical activity (MPA), vigorous-intensity physical activity (VPA), or a combination of them (MVPA) is recommended in order to gain specific health benefits. Application of PA recommendations using fixed absolute intensities (e.g., MPA: 3–6 METs) [22] may lead to insufficient health benefits in our group of pregnant and nonpregnant women. In our sample, activity according to fixed absolute moderate intensity may be not intense enough to provoke larger adaptions of the cardiorespiratory system since 3 METs were lower compared to the intensity at VT_1_ in all women except four. On the contrary, in individuals with lower fitness level, overloading or discouragement due to unattainable recommendations could be the result of recommendations based on absolute intensities [10]. Individually tailored metabolic or physiological accelerometer cut points were already shown to reduce this methodological error and to provide more meaningful results [19,20]. To determine individualized accelerometer cut points, a three-phase model [16] can be applied which allows one to detect the transition from phase 1 to phase 2 of energy supply, independent of the individual performance level, by using an individual threshold like VT_1_. Phase 1 is characterized by a metabolically inter- and intramuscular balanced situation. Activities within this phase can be maintained for several hours without becoming fatigued [23]. The metabolic situation in phase 2 is systemically balanced but activity duration is limited [16]. Deliberate activity in phase 1 or 2 will therefore cause specific adaptions on a local and systemic level and the exact definition of these phases enables precise prescription and interpretation of intensity [14,24]. Therefore, optimized health benefits are suspected when recommendations are based on individual metabolic thresholds (i.e., VT_1_ = lower limit for MPA), which are standard in performance development in structured training processes [25]. Furthermore, a more accurate assessment of the intensity of PA would enable better associations with health outcomes, dose–response relationships, and behavior surveillance [19,26].

The accelerations at absolute cut points (Cp_3.5_, Cp_ind_) were significantly lower compared to VT_1_. Mean values for Cp_3.5_ were higher in pregnant compared to nonpregnant women (e.g., Axivity: 165 ± 118 vs. 95 ± 43 mg). This difference of accelerations between groups was smaller for Cp_ind_ and VT_1_ values as well as for the hip-worn device (Table 2). Such interindividual variability has already been shown for intensity cut points relative to heart rate reserve in a group with heterogeneous cardiorespiratory fitness. A high-fit group had higher accelerometer counts at the same relative intensity compared to a low-fit group [12]. Clear influence on varying accelerometer cut points was also shown for age and overweight/obesity [27,28]. However, age and weight affect cardiorespiratory fitness, which tends to decrease with both age [29] and obesity [30]. In our study, pregnant women had significantly higher body mass and maximum performance capacity was significantly lower compared to nonpregnant women. However, accelerometer cut points were higher (for wrist-worn devices) within pregnant women but showed no correlation with the walking speed at VT_1_. Therefore, accelerometer values from wrist-worn devices could not be attributed to intensity in pregnant women. As walking economics were shown to change in pregnancy [31], higher accelerations in pregnant woman might more likely show differences in walking style. In nonpregnant women, high interindividual variability at the single cut points can be explained by differences in performance capacity, due to walking speeds at VT_1_ (higher speed implicates a higher performance capacity) being significantly correlated to cut points, except for the wrist-worn ActiGraph. Accelerations in wrist-worn devices generally varied between the constant speed increments of IWT and were in some cases generally higher from the start of IWT compared to average values. In contrast, the hip-worn device showed less variability and significant correlations between walking speed and accelerations at VT_1_ (r_preg_ = 0.62/r_non_ = 0.69) in pregnant and nonpregnant women, which is in line with a study by Ozemek et al. [12]. The hip-worn device seems to be less affected by the walking style and may therefore provide more meaningful results, especially in pregnant women. Mean ENMO accelerometer values in nonpregnant women for Cp_3.5_ were in line with recent findings from the literature, presenting similar values for wrist-worn GENEActive (93.2 mg) and ActiGraph (100.6 mg) and hip-worn ActiGraph (69.1 mg) devices [5].

Mean speed at VT_1_ in our study (pregnant: 5.7 km·h^−1^ and nonpregnant: 6.2 km·h^−1^) is comparable to other studies, which determined a walking speed of ≈ 5 km·h^−1^ in older healthy men and women (56 ± 16 yrs) [32] and of 5.1 and 5.5 km/h in postmenopausal women [19,20]. Determination of VT_1_ in a walking test seems to be applicable in a wide range of populations. Furthermore, walking tests are highly practicable due to the short duration of the test (average duration 6.5 min) and the fact that VT_1_ can also be assessed using heart rate variability measurement of a simple heart rate monitor [32]. This enables testing several subjects at once in a short period of time requiring only heart rate monitors and pacing.

In both groups, 1 MET_ind_ was significantly higher compared to the conventional 1 MET value (3.5 mL/kg×min). Higher resting metabolic rates in pregnant women compared to the conventional 1 MET value are different compared to the literature, where no difference to the conventional 1 MET value was found [9]. This might be due to no acclimatization period and the relatively short measurement period of the resting state in our study. Higher resting metabolic rates of our sample of mainly active and well-trained women are in line with the recent literature, where energy expenditure of active healthy women was found to be underestimated by the conventional 1 MET [33]. Determination of the absolute accelerometer cut points at moderate intensity (3 METs) by the MET_ind_ resulted in higher values compared to the conventional equivalent (MET_3.5_), but in significantly different values compared to VT_1_. Therefore, using the MET_ind_ can partly compensate for the error made by using the fixed 1 MET definition, but not for the differences in performance capacity.

However, this study is not without limitations. The protocol of the IWT, with 50 m speed increases, allows a time-efficient determination of the first threshold (average duration 6.5 min). Although small increases of 0.5 km/h per increment favor a fast adaption, 50 m is a relatively short distance to adjust to the pacing speed. However, with increasing speed, walking time of the single increments decreases, which might have increased variance at higher speeds. For the definition of accelerations at a constant speed, increments with longer and equal duration would provide more precise results. Prior to IWT, individual resting metabolic rates were defined from a 15 min supine rest position without any guidance regarding the fasting state. This is not according to the general practice, as usually subjects have to be overnight fasted, run through an acclimating period and a longer measurement period [9,33], which would not have been feasible in pregnant women. Because of this, the determined resting metabolic equivalent needs to be considered with caution. Nevertheless, our resting metabolic equivalent determination was sufficient to get performance parameters related to the subject’s actual metabolic status, considering their actual weight, age, and performance capacity. Furthermore, one could criticize that the tests were conducted by two research groups in different countries. Between groups, we used different gas analyzer devices, possibly influencing the results, although appropriate calibration was performed and devices were shown to provide reliable measurements with adequate validity for field-based measurements [34]. Furthermore, the majority of women in South Africa were black Africans, compared to Caucasian in the Austrian population, which was not considered in the analysis even though African Americans were shown to have lower accelerometer cut points compared to Caucasians in a maximal graded exercise treadmill test [2]. Assessing differences in thresholds, the role of race was considered negligible due to individual analysis. Nevertheless, this approach might have increased interindividual variability of accelerometer data. However, the generalizability of these findings is limited due to the small number of subjects. Future research should take these limitations into consideration and validate the findings in a larger, more representative sample.

## 5. Conclusions

Intensity at absolute 3 MET accelerometer cut points was lower compared to the intensity at VT_1_ independent of the device or placement in pregnant and nonpregnant women. The application of the individual resting metabolic equivalent results in an approximation to the first ventilatory threshold but does not provide an alternative for individually tailored activity cut points. Using absolute accelerometer cut points, which are independent of the individual performance capacity, can lead to different physiological and metabolic strain at the same absolute intensity, possibly causing under- or overloading for a particular person. Therefore, individual thresholds based on physiological parameters, such as the VT_1_, are recommended to quantify the intensity of PA. A short incremental walking test can be a time-efficient method to define these thresholds and, thus, can be used for tailoring accelerometer cut points to individual differences in performance capacity.

## Figures and Tables

**Figure 1 ijerph-17-05651-f001:**
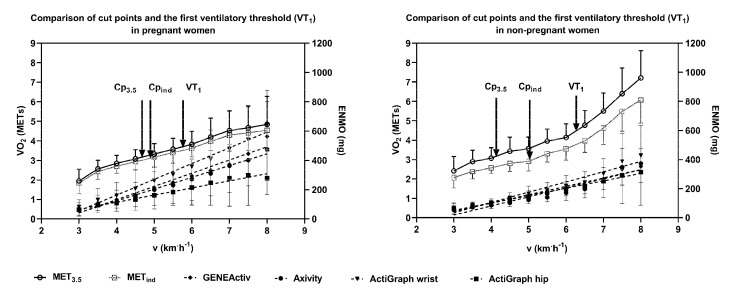
Mean ± SD oxygen uptake (VO_2_) expressed in MET_3.5_ (1 MET = 3.5 mL/kg×min) and MET_ind_ (1 MET = 3.7 ± 0.5/4.2 ± 0.6 mL/kg×min in pregnant/nonpregnant women) as well as the Euclidian norm minus one (ENMO=√(a_x_^2^+a_y_^2^+a_z_^2^)-1g) from GENEActiv, Axivity, ActiGraph wrist and hip for each single load step of the incremental walking test (IWT). Arrows mark cut point values determined from MET_3.5_ (Cp_3.5_), MET_ind_ (Cp_ind_), and the first ventilatory threshold (VT_1_).

**Table 1 ijerph-17-05651-t001:** Comparison of anthropometrics, performance of the incremental walking test (IWT), and the individual resting metabolic equivalent between pregnant and nonpregnant women.

Variables	Pregnant (*n* = 24)	Nonpregnant (*n* = 15)
Age (yrs)	27.7 ± 4.6 ^1^	24.3 ± 2.2
Weight (kg)	69.1 ± 10.6 ^1^	61.2 ± 7.8
BMI (kg/m)	26.5 ± 4.9 ^1^	22.5 ± 3.4
VO_2max_ (mL/kg×min)	18.2 ± 4.4	26.5 ± 8.0 ^1^
VO_2VT1_ (METs/L/min)	3.8 ± 0.7/0.9 ± 0.2	4.9 ± 1.4 ^1^ / 1.0 ± 0.2
v_max_ (km/h)	7.8 ± 0.5	8.3 ± 0.5 ^1^
v_VT1_ (km/h)	5.7 ± 0.5	6.2 ± 0.8 ^1^
MET_ind_ (mL/kg×min)	3.7 ± 0.5 ^2^	4.2 ± 0.6 ^1,2^

BMI = Body Mass Index, VO_2max_ = maximum oxygen uptake, VO_2VT1_ = oxygen uptake at the first ventilatory threshold, v_max_ maximum walking speed, v_VT1_ = walking speed at the first ventilatory threshold, MET_ind_ = calculated individual resting metabolic equivalent, METs = rates of energy expenditure (1 MET is equivalent to 3.5 mL·kg^−1^·min^−1^, results are shown as *M* ± *SD*. ^1^ significantly higher than the comparison group (*p* < 0.05). ^2^ significantly higher compared to conversion 1 MET = 3.5 mL/kg×min (*p* < 0.05).

**Table 2 ijerph-17-05651-t002:** Metric Euclidian norm minus one (ENMO) of the determined accelerometer cut points of the different devices for pregnant and nonpregnant women.

	Pregnant (*n* = 24)			Nonpregnant (*n* = 15)		
Devices	Cp_3.5_ (mg)	Cp_ind_ (mg)	VT_1_ (mg)	Cp_3.5_ (mg)	Cp_ind_ (mg)	VT_1_ (mg)
GENEActiv (wrist)	190 ± 136 ^2^	204 ± 109 ^2^	290 ± 120 ^1^	106 ± 50 ^2^	156 ± 50 ^1,2^	240 ± 85 ^1^
Axivity (wrist)	165 ± 118 ^2^	178 ± 94 ^2^	271 ± 154 ^1^	95 ± 43 ^2^	145 ± 68 ^1,2^	268 ± 204 ^1^
ActiGraph (wrist)	238 ± 171 ^2^	245 ± 142 ^2^	350 ± 166 ^1^	108 ± 58 ^2^	169 ± 58 ^1,2^	276 ± 129 ^1^
ActiGraph (hip)	139 ± 93 ^2^	145 ± 50 ^2^	215 ± 129 ^1^	106 ± 36 ^2^	152 ± 37 ^1,2^	236 ± 85 ^1^

Cp_3.5_ = absolute accelerometer cut point at moderate intensity (3 MET) calculated using the conventional 1 MET = 3.5 m/kg×min value, Cp_ind_ = absolute accelerometer cut point at moderate intensity (3 MET) calculated using the individual resting metabolic equivalent, VT_1_ = first ventilatory threshold, mg = milligravity. ^1^ significantly different compared to Cp_3.5_ (*p* < 0.05). ^2^ significantly different compared to VT_1_ (*p* < 0.0001).

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
