# Peer review of "Absolute Accelerometer-Based Intensity Prescription Compared to Physiological Variables in Pregnant and Nonpregnant Women"

_ijerph, 2020, doi:10.3390/ijerph17165651_

Round 1

Reviewer 1 Report

The topic addressed by the authors of the manuscript is of interest, but one aspect that greatly limits the study is that the references are outdated (54% are over 5 years old), which clearly affects the quality of the manuscript.

A better search for updated information should be made (2015 to 2020), to support the introduction and above all the discussion.

All comments are in the document highlighted in yellow

Introduction

The wording of the text should be improved since the same articles are cited in consecutive paragraphs.

Line 56 to 60: Several studies showed considerable differences between relative 56 accelerometer cut points and their absolute counterpart [10–12]. Estimation of the duration of 57 moderate-to-vigorous physical activity (MVPA) was shorter when applying relative compared to 58 absolute cut points. Independent of the body mass index (BMI), absolute cut points overestimate 59 MVPA compared to relative cut points [10,12].

Line 82 a 92: Two studies by Gil-Rey et al. [20,21] estimated the intensity of PA from individually tailored accelerometer cut points derived from lactate thresholds in postmenopausal women and found similar time for MVPA in high- and low-fit groups. In contrast, MVPA was overestimated in low-fit and more strongly in the high-fit group when using the absolute accelerometer cut points at moderate intensity (3 – 5.9 METs). Individually rather than traditional absolute accelerometer cut points estimate an individual’s activity level (i.e., time spent in different intensities of PA) more accurately. Thus, using individually tailored cut points from physiological thresholds may provoke greater adaptions to exercise, reduced interindividual variability of metabolic responses as well as less  overestimation of PA-intensity [20,21] To date, it is unknown how well absolute accelerometer cut points are related to a physiological threshold such as the VT1 in pregnant as well as in young non-pregnant women.

- It is suggested to finish the introduction with the aim of the study.

Materials y Methods

Line 148. Check wording.The Shapiro-Wilk normality test was used for confirmation of normality. Tip, Change to: The Shapiro-Wilk test was used for confirmation of normality.

Discussion

The wording of the text should be improved since the same articles are cited in consecutive paragraphs.

Line 285 a 292:

 Higher resting metabolic rates of our sample of mainly active and well trained women are in line with the recent literature, where energy expenditure of active healthy women was found to be underestimated by the conventional 1 MET  [35]. Determination of the absolute accelerometer cut points at moderate intensity (3 METs) by the METind resulted in higher values compared to the conventional equivalent (MET3.5), but to significantly different values compared to VT1. Therefore, using the METind can partly compensate for the error made by using the fixed 1 MET definition [35], but not for the differences in performance capacity.

Author Response

Comment 1:

The topic addressed by the authors of the manuscript is of interest, but one aspect that greatly limits the study is that the references are outdated (54% are over 5 years old), which clearly affects the quality of the manuscript.

A better search for updated information should be made (2015 to 2020), to support the introduction and above all the discussion.

Response 1:

Thank you very much for your constructive comment. We are grateful for your suggestion that was very helpful for improving our manuscript. We hope that you will be satisfied with the revised version in which we have updated the references. Therefore, we performed an updated literature search. However, some references could not be updated because they are basic literature in this research field and are still valid or no specific actual literature could be found in “pubmed”. Till know, the relationship between absolute and individual tailored accelerometer cut points were only investigated in a few studies which are included in the manuscript. To the best of our knowledge, our study includes the actual relevant studies regarding the investigated topic.

Comment 2:

The wording of the text should be improved since the same articles are cited in consecutive paragraphs (examples were given).

Response 2:

Following your comment, we revised these paragraphs (marked in the text; Line 56-59, 81-89).

Comment 3:

It is suggested to finish the introduction with the aim of the study.

Response 3:

Following your comment, we changed the order of the sentences accordingly.

Comment 4:

Line 148. Check wording.The Shapiro-Wilk normality test was used for confirmation of normality. Tip, Change to: The Shapiro-Wilk test was used for confirmation of normality.

Response 4:

Thank you very much for that recommendation which helped improving readability. (Line 149)

Comment 5:

The wording of the text should be improved since the same articles are cited in consecutive paragraphs. (285-292)

Response 5:

Following your comment, we revised this paragraph (marked in the text; Line 286-294).

Comment 6:

Conclusion is not supported by the results – must be improved.

Response 6:

We totally agree and deleted one sentence speculation on possible health effects.

If you should have any further recommendations for improving our manuscript, please communicate these to us.

Reviewer 2 Report

The authors present the study that is well executed, with the well-conceived protocol for data gathering from subjects. The results are significant and interesting in this area of study.

One place where the readers can he helped is in the use of term "cut points". In some (mostly non-medical) research communities, this term might be considered to be the jargon or imprecise. When first time used, it would help to add a more complete definition of the term. 

This reviewer sees no need to need to do other adjustments to the text. 

Author Response

Comment 1:

One place where the readers can he helped is in the use of term "cut points". In some (mostly non-medical) research communities, this term might be considered to be the jargon or imprecise. When first time used, it would help to add a more complete definition of the term.

Response 1:

Thank you for carefully reading the manuscript and your comments which were very helpful for improving the manuscript. We agree, that the term “cut point” could be described in more detail. Because the term “cut point” is very common in the research field, we used this term according to the literature without a more detailed definition. If you should have further comments for improving our manuscript, please don´t hesitate to communicate these to us.

  • Arvidsson D, Fridolfsson J, Börjesson M, Andersen LB, Ekblom Ö, Dencker M, et al. Re-examination of accelerometer data processing and calibration for the assessment of physical activity intensity. J. Med. Sci. Sports. 2019,29:1442–52.

Reviewer 3 Report

The manuscript "Absolute accelerometer-based intensity prescription compared to physiological variables in pregnant and non-pregnant women" presented to me for review is very interesting. The authors used modern, objective tools. However, they did not avoid methodological mistakes:

Subjects: The study group is too small... In my opinion, the group is too small for statistical inference.

There is to small information about the study group. Authors should include information about:

  • age of participants
  • use Flow Chart Graph
  • what was the number of pregnancy in each of participants?
  • Whether pregnancy is single or multiple?

Measurements

  • what was the methodology for accelerometers? How many days did the participants wear device? How many hours per day? How did you define valid day?
  • why only one accelerometer was on left hip and why on left (not right) hip ?
  • You should decribe all parameters of accelerometry (epochs, axis etc.)
  • Which algorithm to calculate MET's did you choose?

Results and Discussion are good but I write it once more: In my opinion, the group is too small for statistical inference and general conclusions.

Author Response

Comment 1:

Subjects: The study group is too small... In my opinion, the group is too small for statistical inference.

There is to small information about the study group. Authors should include information about:

age of participants

use Flow Chart Graph

what was the number of pregnancy in each of participants?

Whether pregnancy is single or multiple?

Response 1:

Thank you very much for your constructive comments. We are grateful for your suggestions that were very helpful for improving our manuscript. We hope that you will be satisfied with the revised version in which we have incorporated your points. If you should have any further recommendations for improving our manuscript, please communicate these to us.

We agree that the sample size is small. To address your comment, we added this to the limitations section of the study (Lines 317,318). Nevertheless, this study shows the principle, that accelerometer thresholds determined from absolute intensity thresholds (3 MET moderate) more likely underestimate intensity compared to the individual ventilatory threshold in young non-pregnant as well as in young pregnant woman. The small number of participants limits the generalization for both pregnant and non-pregnant groups, however, in our opinion this number is sufficient to prescribe the methodological error of absolute intensity prescription via accelerometer cut points.

Furthermore, the age of the participants has already been included in table 1. We additionally included this information in the results paragraph (Lines 158, 159) for clarification.

Thanks for your suggestion regarding the flow chart. We included some further information about the total number of performed and analyzed tests in the subjects and results paragraph (Lines 100,102;159,160. As we could include all information regarding participant groups in these paragraphs, we would like to avoid to include an additional figure.

We agree that with the reviewer that this detailed information may be interesting, however, as we investigated physiological and accelerometer cut off points we believe that these specific details regarding pregnancy are not directly relevant.

 Comment 2:

Measurements

what was the methodology for accelerometers? How many days did the participants wear device? How many hours per day? How did you define valid day?

why only one accelerometer was on left hip and why on left (not right) hip ?

You should decribe all parameters of accelerometry (epochs, axis etc.)

Which algorithm to calculate MET's did you choose?

Response 2:

Participants only wore the accelerometer during the incremental walking test. From this data we determined the raw accelerations at the absolute 3 MET moderate intensity threshold by using the conventional conversion factor (1 MET = 3.5 mL∙kg-1∙min-1) and applying measured individual resting metabolic rate. (Lines 132-135)

Regarding the wearing of the accelerometers: only the ActiGraph is designed to be worn on the wrist and hip, both other accelerometers (GENEActiv, Axivity) are designed to be worn on wrist (Axivity can also be attached with a patch somewhere else e.g. thigh, but in practice it is worn on the wrist). To standardize the measures wrist worn accelerometers were exclusively applied on the non-dominant arm and hip worn accelerometer on the left side as we did not expect an influence regarding the hip position.

Based on actual research, we used raw accelerometer data expressed as the vector magnitude (mg) calculated using the Euclidian norm minus one in 15 s epochs (Line 139). Comparable to other studies in our opinion we described all necessary parameters.

Round 2

Reviewer 3 Report

Thank you for your review responses.
The authors answered most of the questions or made appropriate corrections.